# Nobiletin Ameliorates Aging of Chicken Ovarian Prehierarchical Follicles by Suppressing Oxidative Stress and Promoting Autophagy

**DOI:** 10.3390/cells13050415

**Published:** 2024-02-27

**Authors:** Jingchun Bai, Xinyu Wang, Yiqiu Chen, Qiongyu Yuan, Zhaoyu Yang, Yuling Mi, Caiqiao Zhang

**Affiliations:** College of Animal Sciences, Zhejiang University, No. 866 Yuhangtang Road, Hangzhou 310058, China; 22117036@zju.edu.cn (J.B.); 22217098@zju.edu.cn (X.W.); 22117122@zju.edu.cn (Y.C.); qiongyu@zju.edu.cn (Q.Y.); 12117050@zju.edu.cn (Z.Y.); yulingmi@zju.edu.cn (Y.M.)

**Keywords:** nobiletin, oxidative stress, mitophagy, ovarian aging, chicken

## Abstract

With the increase in the age of laying chickens, the aging of follicles is accelerated, and the reproductive ability is decreased. Increased oxidative stress and mitochondrial malfunction are indispensable causes of ovarian aging. In this study, the physiological condition of prehierarchical small white follicles (SWFs) was compared between D280 high-producing chickens and D580 aging chickens, and the effect of a plant-derived flavonoid nobiletin (Nob), a natural antioxidant, on senescence of SWFs granulosa cells (SWF-GCs) was investigated. The results showed that Nob treatment activated cell autophagy by activating the AMP-activated protein kinase (AMPK) and Sirtuin-1 (SIRT1) pathways in D-galactose (D-gal)-generated senescent SWF-GCs, restoring the expression of proliferation-related mRNAs and proteins. In addition, the expression of inflammation-related protein NF-κB was significantly enhanced in aging GCs that were induced by D-gal. Nob supplementation significantly increased the antioxidant capacity and decreased the expression of several genes associated with cell apoptosis. Furthermore, Nob promoted activation of PINK1 and Parkin pathways for mitophagy and alleviated mitochondrial edema. Either the AMPK inhibitor dorsomorphin (Compound C) or SIRT1 inhibitor selisistat (EX-527) attenuated the effect of Nob on mitophagy. The protective effect of Nob on natural aging, GC proliferation, and elimination of the beneficial impact on energy regulation of naturally aging ovaries was diminished by inhibition of Nob-mediated autophagy. These data suggest that Nob treatment increases the expression of mitophagy-related proteins (PINK1 and Parkin) via the AMPK/SIRT1 pathways to prevent ovarian aging in the laying chickens.

## 1. Introduction

The ovary plays a vital role in regulating female fertility and is one of the organs most prone to aging. Ovarian function decline is intricately linked to resting follicle depletion, with oocyte aging as a major determinant of fertility decline in females [1,2]. In poultry, it is observed that after 80 weeks of age, there is a notable decline in egg production among laying chickens, accompanied by evident signs of ovarian aging. This aging is characterized by a significant reduction in both the mass and quantity of oocytes, along with an increase in apoptosis among granulosa cells (GCs) in prehierarchical follicles and a rise in the quantity of atretic follicles [3]. Therefore, delaying ovarian senescence can enhance the economic value of laying chickens by maintaining their egg production and overall reproductive capacity [4].

One of the key factors in the aging process is oxidative stress [5]. Loss of tissue and organ function is frequently caused by an excessive buildup of reactive oxygen species (ROS) [6]. Growing evidence underscores the crucial balance between ROS and antioxidant levels within the ovary, which is essential for preserving female reproductive health. Excessive ROS accumulation causes oxidative damage, resulting in damage to oocytes and GCs in the ovaries, a pivotal contributor to fertility decline [7,8]. In rats, studies have demonstrated that oxidative stress can induce conditions like polycystic ovary syndrome and alter ovarian morphology [9]. Moreover, it has been observed that oxidative stress causes apoptosis in bovine GCs in mammals, while activating antioxidant responses can mitigate ovarian GC apoptosis and ameliorate oocyte dysfunction [10]. In poultry, it is noteworthy that the degree of oxidative stress in chickens’ ovaries increases continuously during the aging process, and this is associated with a decrease in fecundity [3]. Therefore, strategies to reduce oxidative stress damage to GCs and enhance antioxidant capacity have emerged as effective methods to mitigate ovarian aging.

Numerous studies indicate that autophagy is necessary for the preservation of redox homeostasis and is also a major cellular defense against oxidative stress [11]. Autophagy serves as a vital cellular defense mechanism, activated by ROS, for adapting to and mitigating oxidative damage by degrading and recycling damaged macromolecules and dysfunctional organelles within cells [12]. In mammals, oocytes have more mitochondria, playing a pivotal role in various aspects of oocyte maturation, fertilization, and early embryonic development. However, as age advances, both the number and activity of mitochondria decrease [13]. Abnormal mitochondrial function is common in many age-related diseases [14]. Mitochondrial dysfunction has been linked to the onset of reproductive senescence, making the improvement of mitochondrial activity a potential target for promoting oocyte regeneration [15]. Meanwhile, mitochondrial failure in GCs, a frequent factor in ovarian aging, leads to an increase in ROS, causing GC apoptosis and harm to ovarian function [16,17]. Therefore, alleviating mitochondrial damage and removing dysfunctional mitochondria have emerged as effective measures to alleviate ovarian aging.

Aging models are invaluable tools for investigating age-related systems. D-galactose (D-gal) is a well-known reducing sugar that has been used in aging models. D-gal was shown to induce the excessive accumulation of advanced glycation products in oxidative metabolism, leading to cell and tissue senescence [18]. Previous studies have demonstrated that in-vitro-cultured ovarian tissues treated with D-gal exhibit significant aging changes in chickens, such as increased levels of oxidative stress and apoptosis [4,19]. The selection of anti-aging compounds and the study of aging mechanisms have extensively used the D-gal-induced senescence model [20]. Recently, many natural plant extracts have been used to alleviate ovarian aging, such as resveratrol [21] and lycopene [3]. Nobiletin (Nob), a polyethoxylated flavonoid derived from citrus fruits, possesses various pharmacological activities, including cell cycle regulation, the inhibition of ROS production, and anti-inflammatory effects [22]. Nob was able to regulate oxidative stress and mitochondrial dysfunction to reduce liver inflammation and cell death [23]. Furthermore, Nob could alleviate hypoxia-induced oxidative stress, reduce ROS and malonaldehyde (MDA) production, and significantly improve mitochondrial dysfunction [24]. Increasing studies have shown that the protective effect of Nob on cells is achieved through the regulation of autophagy and the alleviation of inflammatory response. Nob can alleviate impaired autophagy and mitochondrial dysfunction by activating AMPK and SIRT1/FOXO3a pathway [25]. In mammals, Nob can enhance oocyte quality and developmental competence, increase mitochondrial activity within the oocyte, and reduce ROS content [22]. However, the anti-aging effects of Nob in senescent laying chicken ovaries require further elucidation.

This study aims to investigate Nob’s anti-aging effects on both the D-gal-induced GC premature senescence model and natural aging in chickens. The possible mechanism was explored to reveal the association of Nob’s anti-aging properties with reduced oxidative stress and enhanced mitochondrial autophagy.

## 2. Materials and Methods

### 2.1. Animals and Follicle Collection

Hyline white chickens were grown under conventional husbandry management conditions on a nearby farm. Research complied with the guidelines published by the Guiding Principles for the Care and Use of Laboratory Animals of Zhejiang University (ZJU20220085) regarding the morality of animal experimentation and the use of laboratory animals. Samples were taken from laying chickens in the peak laying period (D280, 280 ± 10 days) and later laying period (D580, 580 ± 10 days). At least five D280 or D580 chickens were used for each experiment. Small white follicles (SWFs, 2–4 mm in diameter) were dissected from the ovaries in a sterile environment and cleaned with precooled phosphate buffered saline (PBS). Isolated SWFs were selected for the in vitro culture, were frozen with liquid nitrogen immediately for subsequent biochemical parameters and Western blot (WB) analysis, or were fixed in paraformaldehyde solution for morphological observation.

### 2.2. Culture of SWFs and Treatments

SWFs from D280 or D580 chicken ovaries were cultured in complete high-glucose DMEM (Hyclone, Tauranga, New Zealand) supplemented with 100 μg/mL streptomycin, 100 IU/mL penicillin (Invitrogen, Carlsbad, CA, USA), and 5% fetal bovine serum (FBS; Hyclone, Tauranga, New Zealand). Each follicle was placed on Millipore filters before being transferred to a 48-well plate with 300 μL complete DMEM. Nob (purity ≥ 98%, J0715BS, Meilunbio, Dalian, China) was first dissolved in dimethyl sulfoxide (DMSO), and then diluted with DMEM. The final concentration of DMSO was not over 0.1%. D-gal powder was directly dissolved in DMEM culture medium, prepared into 1 g/mL stock solution, and then diluted into 2.5, 10, 40 mg/mL solutions for treatment of the cultured SWFs. All cultures were maintained in a humidified atmosphere at 38.5 °C with 5% CO_2_, and the culture medium was replaced every 24 h. After 48 h of treatment, bromodeoxyuridine (BrdU; Sigma-Aldrich, St. Louis, CA, USA) incorporation was conducted at 20 μg/mL for subsequent BrdU immunofluorescence staining, and the proliferating cells could be reflected by BrdU staining. The optional dose of 10 mg/mL D-gal was adopted by examining tissue morphology, cell proliferation, and apoptosis rates. For Nob treatment, SWFs were cultured for 72 h in complete medium supplemented with D-gal (10 mg/mL) and different concentrations of Nob (1, 10, and 100 μg/mL) in combination. The optional dose of 10 μg/mL Nob was determined based on the assessment of tissue morphology and cell proliferation. Dorsomorphin (Compound C (CC, an AMPK inhibitor), HY-13418A, MedChemExpress, Shanghai, China) and selisistat (EX-527, HY-15452, an SIRT1 inhibitor, MedChemExpress, Shanghai, China) were dissolved DMSO and diluted with medium. The final DMSO concentration did not exceed 0.1%. To investigate the effect of Nob on ovarian aging in chickens, naturally aging SWFs (D580) were cultured with Nob (10 μg/mL) for 72 h. There were five SWFs that were randomly selected from the follicle pool in each group with different treatments.

**Experiment** **1.***Establishment of follicle aging model*.

In order to induce senescence, a modified D-gal treatment was adopted [26]. In brief, SWFs from D280 chickens were treated with gradient D-gal concentrations of 2.5–40 mg/mL to induce oxidative damage. According to the evaluation of cell apoptosis rate, a D-gal concentration of 10 mg/mL was selected as the optimal concentration for subsequent experiments.

**Experiment** **2.***Screening of Nob dose*.

SWFs from D280 chickens were treated with Nob with gradient concentrations ranging from 1 to 100 μg/mL. According to the evaluation of cell proliferation and apoptosis rate, 10 μg/mL Nob was adopted as the optimal dose for follow-up experiments.

**Experiment** **3.***Treatment of different inhibitors*.

Before Nob (10 μg/mL) and D-gal (10 mg/mL) treatment, the cultured SWFs were treated with 20 μM CC or 20 μM EX-527 for 24 h, respectively. In the inhibitor treatment group, 20 μM CC or 20 μM EX-527 was administered for 24 h, respectively, and then the SWFs were cultured in complete DMEM for 48 h.

### 2.3. Morphological Observation

For hematoxylin and eosin (H&E) staining, SWFs collected directly from D280 and D580 chickens and SWFs cultured for 72 h were preserved at 4 °C for more than 24 h with 4% polyformaldehyde according to a conventional protocol. Subsequently, they were dehydrated in graded ethanol, cleaned with xylene, embedded in paraffin, sectioned to 5 μm thick slices, and placed on glass slides, and then H&E stained according to standard protocols. Stained sections were then observed using a Nikon Eclipse 80i microscope (Nikon, Tokyo, Japan).

### 2.4. Immunofluorescence Staining

Tissue sections obtained from paraffin-embedded SWFs were deparaffinized and rehydrated. Antigen retrieval was performed using a 10 mM sodium citrate buffer (pH 6.0) for 20 min. Endogenous peroxidase was then blocked with 3% hydrogen peroxide and incubated with 5% goat serum (Boster Biological Technology Co., Ltd., Wuhan, China) for 20 min at room temperature. After washing three times with PBS, the sections were incubated with goat anti-mouse secondary antibody conjugated to TRITC for 1 h at 37 °C in a dark environment. Cell apoptosis was detected using the terminal deoxynucleotidyl transferase-mediated dUTP nick-end labeling (TUNEL) Apoptosis Detection Kit (Vazyme, Nanjing, China) following the manufacturer’s instructions. The nuclei were stained with DAPI (Beyotime, Hangzhou, China) for 10 min. The tissue slides were observed under an IX70 fluorescent microscope, and ImageJ 1.54 software (National Institutes of Health, Bethesda, MD, USA) was used for the analysis.

### 2.5. Western Blot Analysis

SWF tissue was homogenized in precooled RIPA Lysis Buffer supplemented with 1% PMSF (Meilunbio, Dalian, China). The supernatant was aspirated by centrifugation. Protein concentration of the tissues was determined using a BCA kit (A045-4-2, Nanjing Jiancheng Bioengineering Institute, Nanjing, China). Equal amounts of proteins were loaded on and separated by 8–12% SDS-PAGE gel, and then electrotransferred into a polyvinylidene fluoride (PVDF) membrane. After blocking with 5% skimmed milk, the PVDF membranes were incubated overnight with corresponding primary antibodies, including proliferating cell nuclear antigen (PCNA, 1:1000, ab29, Abcam, Cambridge, UK), cyclin D1 (CCND1, 1:1000, ER0722, HUABIO, Hangzhou, China), BAX (1:1000, ER0907, HUABIO, Hangzhou, China), caspase-3 (1:1000, ET1602-39, HUABIO, Hangzhou, China), Bcl-2 (1:1000, ER0602, HUABIO, Hangzhou, China), SIRT1 (1:1000, M1506-3, HUABIO, Hangzhou, China), FOXO3a (1:1000, ER1707-79, HUABIO, Hangzhou, China), NF-κB (1:1000, ER0815, HUABIO, Hangzhou, China), phospho-AMPK (1:1000, AP1002, ABclonal, Wuhan, China), AMPK (1:1000, A1229, ABclonal, Wuhan, China), LC3B PINK1 (1:1000, A0968, ABclonal, Wuhan, China), Parkin (1:1000, A7131, ABclonal, Wuhan, China), and β-actin (1:50000, AC026, ABclonal, Wuhan, China). The secondary antibodies included HRP-conjugated goat anti-rabbit IgG goat polyclonal antibody (1:50000, HA1001, HUABIO, Hangzhou, China) and HRP-conjugated goat anti-mouse IgG goat polyclonal antibody (1:20000, HA1006, HUABIO, Hangzhou, China). After washing with PBST three times, the membranes were treated with the secondary antibodies. Images were captured using a ChemiScope 6200 (Clinx, Shanghai, China). P-AMPK expression was normalized to AMPK, and expression of all other proteins was normalized to β-actin.

### 2.6. RNA Extraction and Quantitative Real-Time PCR

Tissues from cultivated SWFs were processed with Trizol (Invitrogen Co., Carlsbad, CA, USA) for total RNA extraction. Following the manufacturer’s instructions, the HiScript II 1st Strand cDNA Synthesis Kit (Vazyme) was used to reverse transcribe 2 µg of total RNA into cDNA. Real-time fluorescence quantitative PCR (qRT-PCR) was utilized to determine the expression levels of *CDK-2*, *PCNA*, *CCND1*, caspase-3, *Bcl-2*, *CAT*, and *SOD1*. The sequences of primers for PCR analysis are listed in Table 1. All samples were normalized with the average of β-actin using the comparative cycle threshold method (2^−△△Ct^).

### 2.7. Measurement of Oxidative Parameters and ATP Level

Cultured SWFs were collected and homogenized with PBS. Subsequently, centrifugation was performed at 800 rpm at 4 °C for 10 min, and the supernatant was aspirated to obtain a 10% tissue homogenate. The tissue homogenate was used to measure oxidation parameters and determine total protein concentration. The activity of CAT (A00171-1, Nanjing Jiancheng Bioengineering Institute, Nanjing, China) and T-SOD (A001-1-1, Nanjing Jiancheng Bioengineering Institute, Nanjing, China), full antioxidant capacity (T-AOC) (A015-3-1, Nanjing Jiancheng Bioengineering Institute, Nanjing, China), the concentrations of MDA (A003-1-1, Nanjing Jiancheng Bioengineering Institute, Nanjing, China), GSH (A006-2-1, Nanjing Jiancheng Bioengineering Institute, Nanjing, China), and hydrogen peroxide (H_2_O_2_) (A064-1-1, Nanjing Jiancheng Bioengineering Institute, Nanjing, China), and total protein concentration were determined using kits (Nanjing Jiancheng Bioengineering Institute, Nanjing, China) according to the manufacturer’s instructions. Additionally, the cultured SWFs were homogenized in boiling water, heated in boiling water for 10 min, and then subjected to centrifugation. The ATP level was determined by an ATP assay kit (A095-1-1, Nanjing Jiancheng Bioengineering Institute, Nanjing, China), following the instructions provided by the manufacturer.

### 2.8. Transmission Electron Microscopy

Cultured SWF tissue was collected and fixed overnight with 2.5% glutaraldehyde in 0.1 M PBS (pH 7.0). Subsequently, the tissue was fixed with 1% OsO_4_ for 1.5 h, and then washed three times for 15 min at each step with PBS. The sample was dehydrated using a graded series of ethanol and acetone. Finally, it was embedded, sectioned into ultrathin slices, and stained. The SWF tissue samples were observed with a Hitachi Model H-7650 (Hitachi, Ibaraki, Japan) transmission electron microscope (TEM).

### 2.9. Statistical Analysis

The experiments were conducted three times. Data analysis was performed using GraphPad Prism9 software, employing one-way ANOVA, followed by either Tukey’s or Dunnett’s post hoc test. Statistical significance was determined when *p* < 0.05.

## 3. Results

### 3.1. Morphological, Proliferative, Apoptotic, and Laying Performance Alterations Associated with Aging in SWFs

H&E staining of SWFs from D280 and D580 chickens revealed distinct morphological differences. In D280 SWFs, GCs were densely arranged, and the theca layer (TL) and granulosa layer (GL) showed a distinct boundary. In contrast, the follicular structure of D580 SWFs exhibited a looser arrangement of the GL and TL with incompact cell distribution (Figure 1A). The proliferation of GCs in D280 chicken SWFs was much higher than in D580, according to the BrdU staining experiment (Figure 1C). Meanwhile, compared with D280, D580 SWFs had a significantly higher number of TUNEL-positive GCs (Figure 1D). Ten healthy laying chickens from each group were randomly selected for preovulatory follicle development analysis. The results verified a significant decrease in hierarchical follicles in laying chickens from D280 to D580 (Figure 1B).

### 3.2. Establishment of the SWF Aging Model

To determine the optimal concentration of D-gal for establishing the aging model, SWFs from D280 chickens were cultured with a range of D-gal concentrations for 72 h. The morphology of the cultured SWFs was observed by H&E staining. The findings indicated that as D-gal concentration increased, the arrangement of GCs gradually became loose and disordered (Figure 2A). Additionally, the TUNEL assay confirmed a significant increase in the apoptosis of GCs with the rising D-gal concentration. Remarkably, the degree of apoptosis in the GC layer of D280 SWFs exposed to 10 mg/mL D-gal (Figure 2B,C) was comparable to that of D580 (Figure 1D). Therefore, the ideal concentration of D-gal to induce senescence in SWF-GCs was determined to be 10 mg/mL.

### 3.3. Effect of Different Concentrations of Nob on D-Gal-Induced Senescence of SWFs

To assess the potential reversal of D-gal-induced aging in SWFs, Nob was applied as a natural plant extract. After 72 h of treatment with D-gal, a decrease in BrdU labeling rates in GCs was observed, indicating reduced cell proliferation. Nob treatment significantly restored the reduced GC proliferation caused by D-gal exposure (Figure 3A,B). Similarly, the TUNEL assay results demonstrated that Nob significantly reduced the increased proportion of TUNEL probe-labeled cells after D-gal treatment (Figure 3C,D). Correspondingly, qRT-PCR analysis revealed that Nob increased expression of the genes *PCNA* and *CDK-2* related to proliferation, and reversed the expression levels of the genes related to apoptosis, including a decrease in caspase-3 gene level and an increase in *Bcl-2* gene level, which D-gal had induced in SWF-GCs of D280 chickens (Figure 3E–H). Western blot experiments further confirmed that D-gal treatment decreased PCNA and Bcl-2 expression while increasing BAX expression. After supplementing with Nob, the expression of PCNA and Bcl-2 significantly increased, and the presentation of BAX significantly decreased (Figure 3I,J). The optimal concentration for reversing GC senescence was determined to be 10 µg/mL Nob and was used for subsequent experiments.

### 3.4. Effects of Nob on the D-Gal-Induced Aging in SWFs

An in vitro aging model for SWFs was established by treating them with 10 mg/mL D-gal for 72 h. H&E staining revealed changes in the shape of growing follicles caused by D-gal treatment, and combined therapy with Nob successfully reversed the damage induced by D-gal in SWF-GCs. Notably, Nob treatment alone had no significant effect on growing follicles (Figure 4A). Furthermore, an incubation experiment with BrdU showed a considerable increase in the rate of BrdU labeling in SWFs subjected to combined Nob treatment compared to the D-gal treatment group (Figure 4B,C). Nob effectively prevented the increase in the TUNEL labeling rate induced by D-gal (Figure 4D,E). The rates of BrdU and TUNEL labeling remained unchanged with Nob treatment alone. Western blotting analysis demonstrated that D-gal observably decreased the expression of PCNA and CCND1. However, simultaneous supplementation with Nob reversed these alterations, significantly rescuing the decrease in CCND1 expression induced by D-gal. When treated with Nob alone, the protein expression of these markers was slightly higher than in the control group. In addition, Western blot analysis of apoptosis-related proteins showed a significant increase in BAX and caspase-3 expression in the D-gal-induced aging SWFs compared to the control group. Consistent with expectations, BAX and caspase-3 expression was reduced by simultaneous supplementation with Nob. Meanwhile, treatment with Nob alone significantly decreased BAX and caspase-3 expression (Figure 4F–J).

### 3.5. Effect of Nob on D-Gal-Induced Aging SWFs’ Protection and Decreased Antioxidant Capacity

To further elucidate the impact of Nob on aging SWFs induced by D-gal, we examined the gene expression related to proliferation and apoptosis, as well as antioxidant enzyme activity, MDA, and hydrogen peroxide (H_2_O_2_) content in the four groups. The results revealed that the mRNA levels of *CDK-2*, *CCND1*, *Bcl-2*, *CAT*, and *SOD* decreased sharply after D-gal treatment. However, combined culture with Nob for 72 h recovered expression of these genes to a similar level as the control. Meanwhile, RT-qPCR detection revealed that Nob decreased the elevated expression of D-gal-induced apoptosis-related gene caspase-3 (Figure 5A–F). The transcript levels of each gene were similar between the Nob alone group and the control group. The results also manifested that the total antioxidant capacity (T-AOC) and the glutathione (GSH) contents in SWFs induced by D-gal were observably lower than those in the control groups (Figure 5G,H). After being exposed to D-gal for 72 h, both catalase (CAT) and total superoxide dismutase (T-SOD) activity were dramatically decreased (Figure 5I,J). However, adding Nob effectively restored these decreased changes induced by D-gal injury. Nob treatment alone did not have a marked effect on these levels (Figure 5E–J). Meanwhile, the levels of H_2_O_2_ and MDA in the aged SWFs induced by D-gal were also significantly increased, while simultaneous administration with Nob could inhibit the above increases. Treatment with Nob alone for 72 h considerably reduced the levels of H_2_O_2_ and MDA in the D-gal-induced aging SWFs (Figure 5K,L). These results suggested that Nob supplementation can prevent the decline of the antioxidant capacity of D-gal-induced aging SWFs.

### 3.6. Effect of Nob on Down-Regulation of AMPK and SIRT1 Pathways in the D-Gal-Induced Senescent SWFs

TEM results showed that D-gal induced mitochondrial swelling and apparent damage in senescent SWF-GCs. After 72 h of concurrent Nob and D-gal treatment, the mitochondrial swelling of living GCs was reduced. Meanwhile, there was no discernible change in the mitochondrial morphology between the Nob-treated group and the control group (Figure 6A). These results suggested that Nob can partially repair the ultrastructural damage to living particles caused by D-gal. Western blotting demonstrated that compared to the D-gal treatment and control group, the phosphorylation and activation of AMPK, a key regulator of mitophagy, were significantly increased in the simultaneous treatment with Nob and Nob treatment alone groups. Meanwhile, in D-gal-induced senescent SWFs, the expression levels of SIRT1 and FOXO3a were decreased considerably, while the autophagy marker LC3B was slightly increased. The addition of Nob effectively prevented the decrease in SIRT1 and FOXO3a expression and significantly increased the expression of LC3B (Figure 6C–F). To confirm the protective effect of Nob on mitophagy, we examined the protein expression levels of PINK1 and Parkin, critical components of the mitophagy pathway. The groups treated with Nob showed considerably higher expression levels of PINK1 and Parkin compared to the control and D-gal groups. As expected, the protein expression of NF-κB in senescent SWFs induced by D-gal was considerably higher than that in the control group, and the simultaneous administration with Nob effectively inhibited this increase. After Nob alone treatment for 72 h, the expression of NF-κB was markedly lower than that in the control group (Figure 6G–J). Meanwhile, co-culture with Nob significantly increased the ATP content in D-gal-induced senescent SWFs, and the ATP content in the treatment with Nob alone was marginally higher than that of the control group (Figure 6L).

### 3.7. Nob Attenuates the Mitochondrial Damage Induced by D-Gal in SWF-GCs through the Activation of Mitophagy via AMPK and SIRT1 Pathways

To determine the involvement of the AMPK and SIRT1 pathways in Nob-induced mitophagy, specific inhibitors were utilized to block the expression of these two proteins individually. To test this, the AMPK/SIRT1 signaling pathway was blocked using CC and EX-527. The results showed that CC treatment significantly reduced the protein expression of AMPK, p-AMPK, SIRT1, FOXO3a, and LC3B in simultaneous administration with Nob (Figure 7A–E). In addition, after the addition of EX-527, the expression of SIRT1 was significantly inhibited, and p-AMPK/AMPK, FOXO3a, and LC3B expression was also significantly decreased, consistent with the expectation that SIRT1 acted synergistically with AMPK to activate FOXO3a (Figure 7J–N). Notably, the specific AMPK inhibitor CC eliminated the enhanced influence of Nob on the expression of PINK1 and Parkin, two downstream mitophagy pathways (Figure 7F–H). Similarly, adding the specific SIRT1 inhibitor, EX-527, yielded the same results (Figure 7O–Q). Moreover, the increase in ATP content in senescent SWFs induced by D-gal treated with Nob was significantly inhibited by adding AMPK and SIRT1 inhibitors, respectively (Figure 7I,R).

### 3.8. The Effect of Nob on Delaying Natural Ovarian Aging Is Achieved by Activating Mitophagy

To evaluate whether Nob treatment could protect naturally aging SWFs by activating mitophagy, two groups of D280 and D580 SWFs were cultured in vitro. These groups consisted of a Nob-treated group and a blank control group, both cultured for 72 h. The morphology of the cultivated SWFs was observed using H&E staining. The results demonstrated that Nob treatment increased follicle development in the D580 group and preserved the morphology of developing follicles in a state similar to that in the D280 group, with a tight arrangement of the granulosa layer and membrane layer. There were no significant differences between the D280 Nob-cultivated group and the blank control group. In contrast, the untreated D580 group showed apparent GC damage, and the granulosa layers were arranged irregularly, loosely, and separated from the membrane layers (Figure 8A). Moreover, through the BrdU incubation experiment, we found that the rate of BrdU labeling in Nob-treated SWFs of D580 was significantly higher than in the untreated D580 group. The Nob-treated SWFs of D280 did not vary substantially from the control group regarding the BrdU-labeling rate (Figure 8B,C). The results of the TUNEL assay revealed a significant reduction in the proportion of TUNEL-positive cells in naturally aging SWFs following treatment with Nob. At the same time, Nob supplementation had no discernible effect on the number of TUNEL-positive cells in D280 SWFs (Figure 8D,E). These findings suggest that Nob supplementation effectively delays the natural aging-related apoptosis of SWFs. The results of Western blot analysis showed that Nob treatment significantly up-regulated the PINK1 and Parkin expression in both D280 and D580 SWFs while significantly decreasing NF-κB expression in SWFs of D580 (Figure 8F–I). PCNA and CCND1 expression levels increased dramatically in the D280 and D580 SWFs following 72 h in vitro treatment with Nob. In addition, Nob supplementation significantly down-regulated BAX expression in D580 SWFs (Figure 8J–M). These results suggest that Nob supplementation can alleviate the inflammatory response during aging by activating mitophagy, thereby maintaining the homeostasis of SWF-GC proliferation and apoptosis during senescence.

## 4. Discussion

The ovaries of female animals, pivotal reproductive organs, are highly susceptible to the aging process. Ovarian aging is characterized by delayed follicular development, decreased number of oocytes, and depletion of the pregraded follicular pools. The deterioration of ovarian function is significantly impacted by the excessive accumulation of senescent organelles in follicular GCs. Meanwhile, mitochondrial dysfunction is an essential indicator of cellular senescence [27]. In this investigation, we discovered that Nob supplementation extensively promoted proliferation and cell autophagy in D-gal-induced senescent SWF-GCs, thereby delaying ovarian aging. Our findings suggest that the protective effects of Nob are mediated by activating AMPK and SIRT1 pathways and enhancing the expression of FOXO3a longevity genes. Furthermore, Nob could up-regulate the expression of the PINK1–Parkin pathway, reduce the expression of inflammatory protein NF-κB, and promote the renewal of aging mitochondria (Figure 9). These findings provide novel insights into the underlying processes that underlie Nob’s anti-aging properties.

In the normal development of follicles, a dynamic balance is maintained between the cellular antioxidant capability and reactive oxygen species generation [28]. However, as age advances, the gradual accumulation of ROS in follicle GCs leads to oxidative stress, accompanied by a decline of antioxidant capacity-induced ovarian senescence [29]. Evidence showed that GCs taken from old cow antral follicles have significantly reduced cell proliferation activity and mitochondrial number compared with young cows [30]. In this study, the quantity of follicles during the peak laying period (D280) was significantly higher than that of late laying (D580) in laying chickens. Meanwhile, the arrangement of granulosa cell layers in SWFs changed from a tight and regular structure to a loose and disordered one with increasing age. Notably, the results of BrdU experiments indicated a significant decrease in the proliferation of GCs within the D280–D580 period, while TUNEL experiments revealed considerably more apoptosis in the D580 group than in the D280 group. Concurrently, with the progression of ovarian aging, the turnover rate of organelles decreased significantly, leading to mitochondrial dysfunction [31]. This resulted in a decline in oocyte quality and a reduction in the number of active mitochondria in GCs [32]. Similar to mammals, aging is an essential factor in laying chickens’ declining egg production, accompanied by a decline in ovarian antioxidant capacity [4] and the onset of mitochondrial dysfunction occurs at the late oviposition stage. In contrast, NF-κB expression was considerably higher in the D580 group than in the D280 group, suggesting that the decline in the reproductive ability of laying chickens is linked to the extent of mitophagy. The functional lifespan of laying chickens is directly related to the rate of ovarian aging [3]. Our study discovered that PINK1–Parkin expression in D580 chickens was slightly higher than in D280 chickens. In comparison, NF-κB expression was considerably higher in the D580 group than that in the D280 group, indicating that the decline in the reproductive ability of laying chickens is related to the power of mitophagy. Therefore, the regulation of mitophagy and anti-inflammation may be a feasible way to delay ovarian aging.

Increasing evidence suggests that supplementation of plant extracts, antioxidants, and autophagy induction are effective measures to delay ovarian aging [33]. With anti-cancer, anti-inflammatory, and antioxidation activities, Nob is a polyethoxylated flavonoid commonly prevalent in citrus fruit peels [34]. A previous study has shown that Nob can ameliorate hepatic ischemic and reperfusion damage by promoting the SIRT1/FOXO3a pathway, thereby alleviating mitochondrial biogenesis and autophagy [25]. Moreover, Nob has been found to induce ferroptosis in human melanoma cells by controlling the GSK3 beta-mediated Keap1/Nrf2/HO-1 signaling pathways, which are critical in inhibiting tumor growth [35]. Previous research has reported that Nob could alleviate the adverse effects of AKT inhibitors on the PI3K/AKT pathway to regulate cell proliferation and differentiation [36]. This intervention was shown to have a positive impact on in vitro development of preimplantation bovine embryos and the overall quality of bovine embryo development [37]. However, the mechanism of the anti-aging action of Nob in the follicles of 580-day-old chickens at the end of laying still requires elucidation. The ovary, vital for reproductive and endocrine functions, is more vulnerable to aging than other organs. The primary characteristic of ovarian aging is a gradual decline in oocyte quality and quantity [38], accompanied by oxidative stress, apoptosis, telomere disruption, and mitochondrial dysfunction [39]. In a state of cell senescence where the cell cycle is stagnant, SIRT1, PCNA, CCND1, and Bcl-2 expression decreases, while BAX, caspase-3, and NF-κB expression increases [40]. In this study, senescent SWFs induced by D-gal exhibited nearly all of the characteristics mentioned above, which was consistent with the results of previous research [3], suggesting that the construction of our model was successful. Building on this foundation, this study investigated the protective impact of Nob on aged SWF tissues induced by D-gal. In addition, it was demonstrated that Nob had a specific anti-aging effect on naturally aging follicles. Previous evidence has indicated that after the addition of FSH, the proportion of β-galactosidase-positive cells decreased by 45% compared to the D-gal group [18]. Lycopene supplementation could restore the proliferation of D-gal-induced senescent cells by 73% and reduce the apoptosis rate by 67% [3]. In this study, co-culture with Nob increased the proliferation rate by 60% and decreased the apoptosis rate by 75% in senescent SWFs induced by D-gal. These results provide compelling evidence that Nob can alleviate premature senescence in SWF-GCs of laying chickens. However, the underlying mechanisms require further investigation.

Oxidative stress significantly contributes to severe ovarian dysfunction [41]. Markers such as MDA, H_2_O_2_, GSH, CAT, T-AOC, and T-SOD expression are closely correlated with age. With aging, ovarian oxidative stress increases and antioxidant status decreases [21]. The ovarian aging model induced by D-gal is widely used to examine the mechanism of premature ovarian failure (POF). Patients with POF exhibit markedly reduced ovarian function, increased GC apoptosis, oxidative damage, and endoplasmic reticulum (ER) stress [41]. Therefore, the diminished quality of aging follicle GCs is connected to the generation of oxidative damage, highlighting the potential for reducing oxidative damage to SWF-GCs in postponing ovarian aging [42]. In this study, we found that the levels of MDA and H_2_O_2_ were much higher in the senescent SWF tissues induced by D-gal compared to the control group. Meanwhile, T-AOC, GSH, CAT, and T-SOD activities were dramatically reduced. In addition, the morphology of GCs of SWFs was dramatically damaged after treatment with D-gal, which could be significantly alleviated by simultaneous Nob supplementation.

Dysfunction of autophagy often occurs with aging, and its activation plays a pivotal role in the removal of ROS. Enhanced autophagy can delay cellular senescence, which makes autophagy central to the aging process [43]. Ovarian aging is characterized by quantitative and qualitative changes in the oocyte reserve and mitochondrial dysfunction. In the natural development of follicles, mitochondria are essential centers of energy metabolism in cells [44]. During the aging process, the excess production of ROS results from the accumulation of dysfunctional mitochondria and impaired oxidative phosphorylation. This disrupts the homeostasis of the oxidative stress response network and further aggravates cell damage. Numerous aging-related disorders may be exacerbated by mitochondrial dysfunction and insufficient autophagy [45]. Mitophagy and autophagy are processes responsible for removing damaged mitochondria and are considered as having positive roles in anti-aging [46]. Therefore, the activation of mitophagy can prevent premature ovarian failure by reducing mitochondrial damage in SWFs. In our study, treatment with D-gal led to mitochondrial dysfunction, fragmentation, and swelling in the SWF-GCs of laying chickens. However, simultaneous treatment with Nob and D-gal for 72 h alleviated the mitochondrial swelling in GCs. At the same time, between the treatment group receiving only Nob and the control group, there was no discernible alteration in mitochondrial morphology, suggesting that Nob has a specific alleviation effect on the ultrastructural damage of granules caused by D-gal. The studies show that Nob supplementation can restore the impaired autophagic flux, prevent and treat acute myocardial infarction, and protect the myocardium in rats [47]. It can also improve the function of autophagy, reduce mitochondrial injury, further improve the function of mitochondria, increase ATP production, decrease ROS production, inhibit inflammation, and prevent apoptosis and aging [48]. Our experiments align with these findings, as Nob co-culture with SWFs enhanced the extent of autophagy following D-gal stimulation. This was supported by the significantly higher expression of LC3B and ATP content in D-gal-induced senescent SWFs after the addition of Nob compared to the D-gal group.

There is growing evidence that autophagy is crucial for many mechanisms mediating life extension, such as caloric restriction [49]. Researchers have embarked on a quest to discover direct autophagy modulators to delay aging and enhance overall health and lifespan [50]. AMPK is one of the core regulators of eukaryotic cell biology and metabolic regulation, making it a critical control valve for metabolism and mitochondrial homeostasis [51]. AMPK is activated in response to a decrease in intracellular ATP production [52]. Studies have shown that phosphorylation of AMPK can further regulate autophagy protein (LC3), prevent apoptosis and inflammation, and alleviate aging [53]. SIRT1 is the most widely and prominently researched member of sirtuins and is strongly associated with regulating inflammation and longevity. Studies have highlighted the critical function of SIRT1 in triggering mitophagy and engulfing damaged mitochondria [54]. In addition, it serves as an essential target related to apoptosis and oxidative stress [55]. AMPK and SIRT1 are tightly linked in the regulation of autophagy, energy metabolism, and anti-oxidative stress, and often act synergistically to regulate their expression [56]. AMPK serves as both an activator of SIRT1 and a recipient of its counter-regulation [57]. The SIRT1/FOXO3a pathway is a classical route to activate autophagy and regulate oxidative stress. Previous studies showed that Nob can alleviate liver ischemia and reperfusion injury via triggering SIRT1/FOXO3a-mediated autophagy and the function of mitochondria [26]. Meanwhile, the activation of FOXO3a could promote PINK1–Parkin expression, activate mitochondrial phagocytosis, and inhibit cell apoptosis through E3-ubiquitin ligase [57,58]. In our study, Nob supplementation led to a marked increase in p-AMPK expression in senescent SWFs induced by D-gal. The highest p-AMPK expression was observed in the group supplemented with Nob alone, indicating a substantial elevation in AMPK activation in senescent SWFs following Nob supplementation. Meanwhile, the expression of SIRT1 and FOXO3a in senescent SWFs induced by D-gal with Nob was also significantly increased. Our findings revealed that the effects of Nob on autophagy and mitophagy were reduced by blocking the AMPK and SIRT1 pathways, as shown by decreased protein expression of LC3B and PINK1–Parkin pathways, as well as ATP levels. These results suggest that the AMPK/SIRT1–FOXO3a pathway plays a crucial role in the relationship between Nob and mitophagy. This indicates that Nob promotes autophagy and the removal of damaged mitochondrial by activating the AMPK/SIRT1 pathway, thereby delaying ovarian aging in laying chickens. This result provided a significant perspective of Nob in application to poultry production and needs to be verified by further in vivo feeding trials.

## 5. Conclusions

In conclusion, Nob has demonstrated its capacity to alleviate cell damage and elevate mitochondria activity in both D-gal-induced aging and naturally aging ovarian follicles via regulating autophagy in mitochondrial homeostasis. Therefore, targeting the AMPK/SIRT1–FOXO3a pathway by Nob may be a valuable measure to delay ovarian aging in laying chickens after peak laying.

## Figures and Tables

**Figure 1 cells-13-00415-f001:**
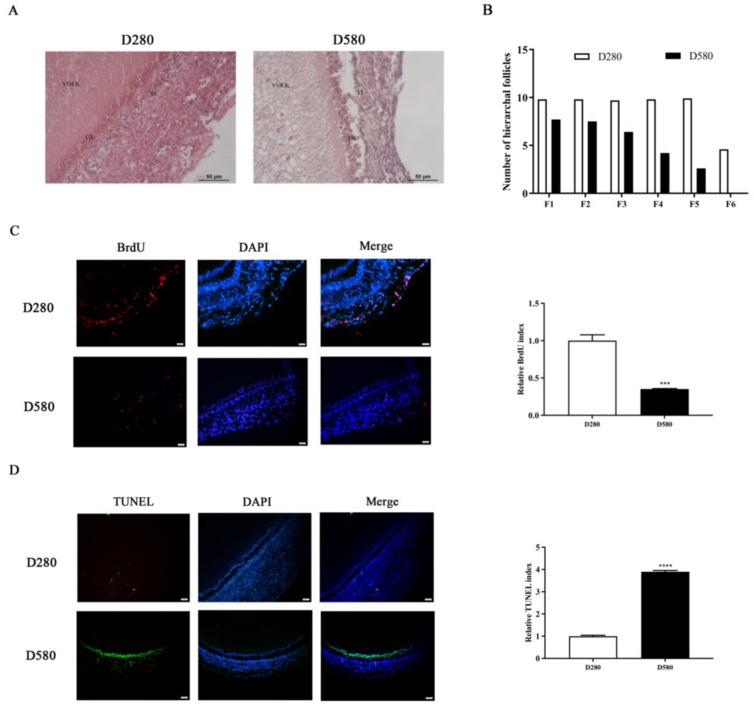
Follicular histology, proliferation, apoptosis, and the number of preovulatory follicles (F1–F6) in D280 and D580 chickens. (**A**) Follicular morphology in the SWFs of chickens aged 280 and 580 days. Scale bar: 50 µm. (**B**) The numbers of preovulatory follicles were compared between the D280 and D580 chickens (n = 10). (**C**,**D**) Immunofluorescent staining to label the proliferating cells with BrdU (red) and apoptotic cells with TUNEL (green) within SWFs. The nuclei were stained with DAPI (blue). Scale bar: 50 µm. *** *p* < 0.001, **** *p* < 0.0001 represent the difference between the groups in the SWF-GCs of chickens aged 280 and 580 days.

**Figure 2 cells-13-00415-f002:**
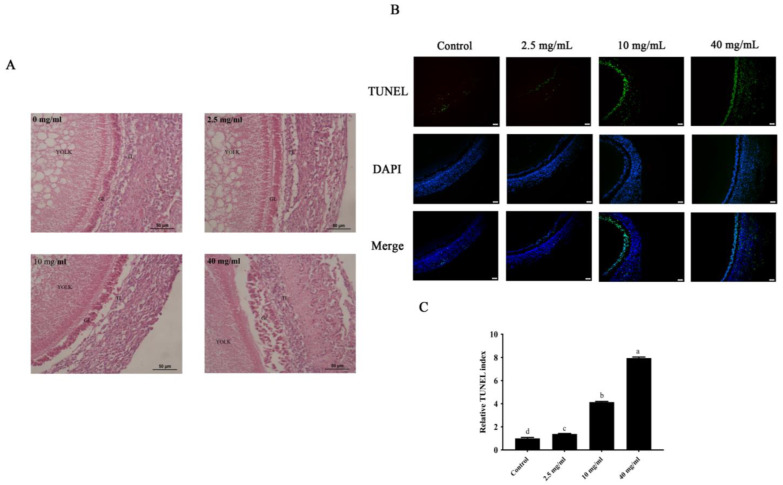
Treatment with D-gal established an aging model in SWFs from D280 chickens. (**A**) The SWFs from D280 chickens were stained with H&E after being treated with D-gal at concentrations of 0, 2.5, 10, and 40 mg/mL for 72 h. (**B**,**C**) SWFs were labeled with TUNEL (green) as indicated. The nuclei were stained with DAPI (blue). Scale bar: 50 µm. Different lowercase letters indicate statistically significant differences (*p* < 0.05).

**Figure 3 cells-13-00415-f003:**
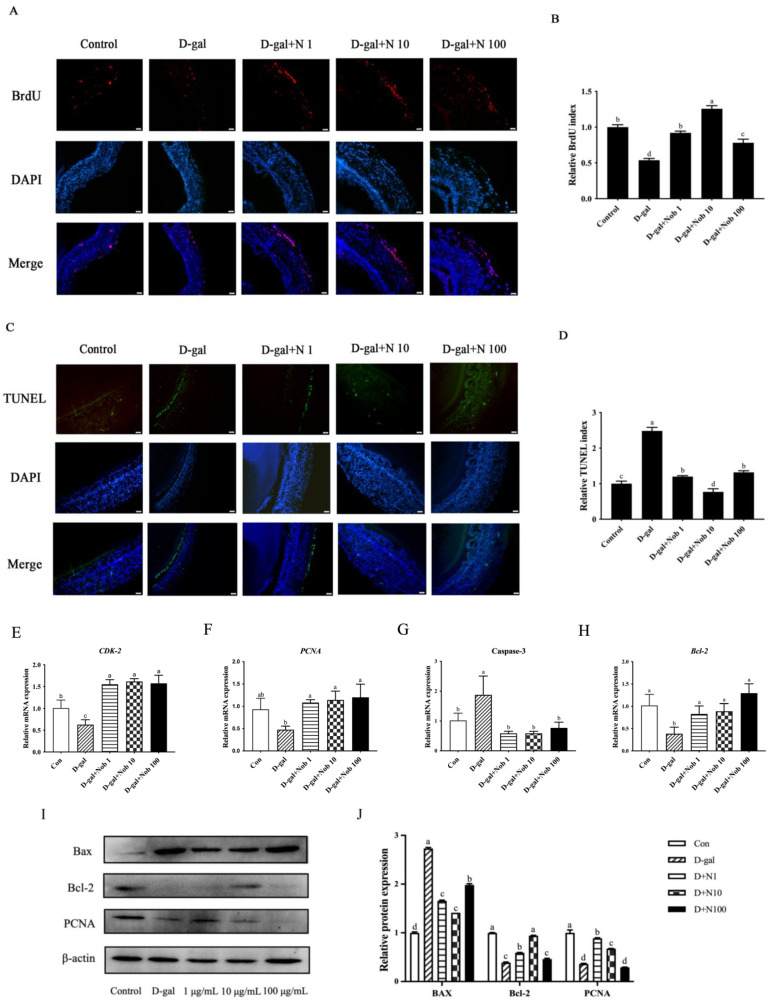
The effect of Nob on the decreased proliferation and increased apoptosis of SWF-GCs in D280 chickens induced by D-gal. (**A**–**D**) SWF-GCs from D280 chickens were treated with D-gal (10 mg/mL) and cultured with Nob at 1, 10, and 100 µg/mL concentrations for 72 h. SWFs were incubated with BrdU (red) and labeled with TUNEL (green) as indicated. The nuclei were stained with DAPI (blue). Scale bar: 50 µm. (**E**–**H**) Transcription levels of PCNA, CDK-2, caspase-3, and Bcl-2 in SWFs after 72 h culture treatment. (**I**,**J**) BAX, Bcl-2, and PCNA expression levels in cultured SWFs were determined by Western blotting, with β-actin as the control. Different lowercase letters indicate statistically significant differences (*p* < 0.05).

**Figure 4 cells-13-00415-f004:**
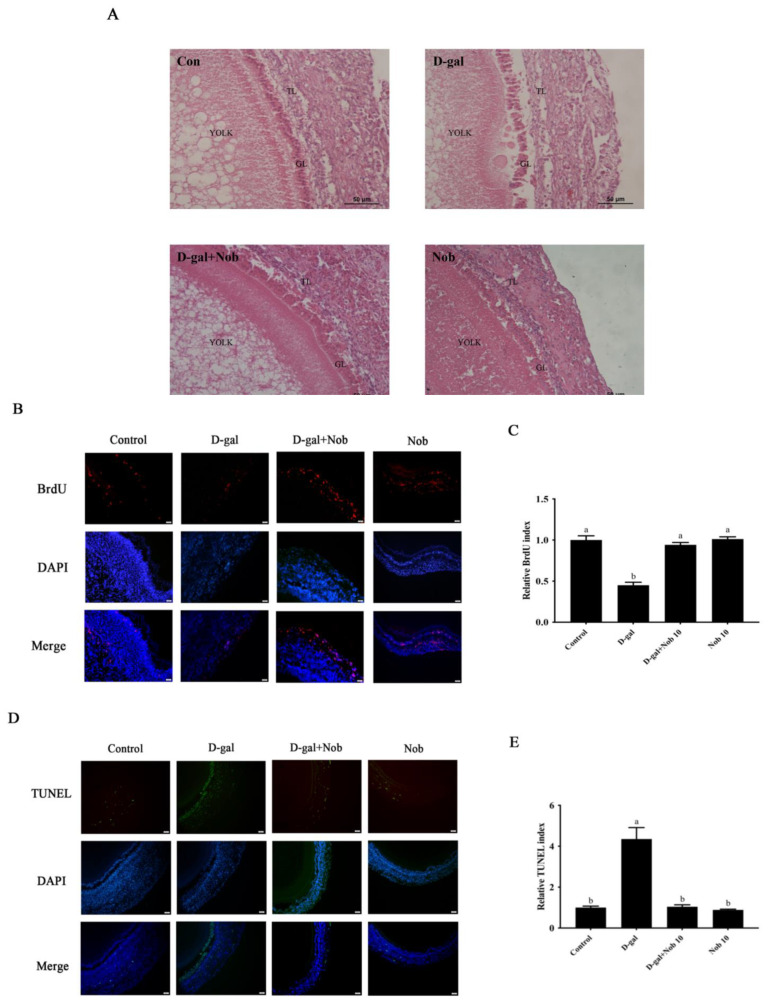
Protective effects of Nob on D-gal-induced aging SWFs. (**A**) The impact of Nob on D−gal−induced morphological alterations in SWFs. (**B**,**C**) The effect of Nob on the decrease in BrdU (red) index induced by D−gal. (**D**,**E**) Effect of Nob on D-gal-rendered TUNEL (green) index elevation. The nuclei were stained with DAPI (blue). Scale bar: 50 µm. (**F**–**J**) Relative expression of proteins related to cell proliferation and apoptosis, β-actin as the control. Different lowercase letters indicate statistically significant differences (*p* < 0.05).

**Figure 5 cells-13-00415-f005:**
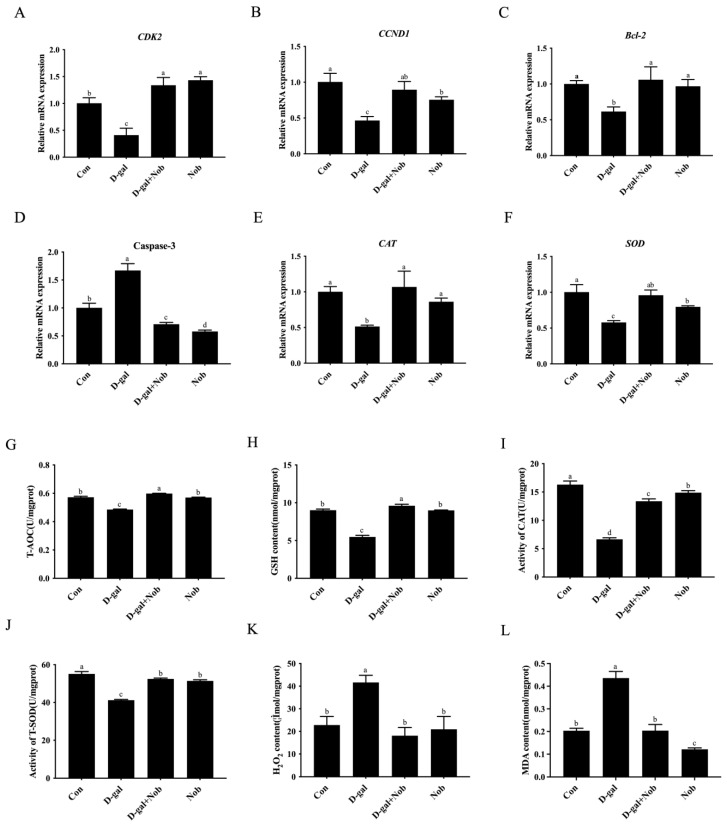
Effects of Nob on decreased expression of genes associated with apoptosis, proliferation, and antioxidant status in D-gal-induced aging SWFs. (**A**–**D**) The genes for proliferation and apoptosis in SWFs, CDK-2, CCND1, Bcl-2, and caspase-3, were identified by RT-qPCR, and β-actin was chosen as the internal control. (**E**,**F**) The antioxidation-related genes, CAT and SOD, were detected by RT-qPCR, and β-actin was used as an internal control. (**G**–**L**) Effect of Nob on decreased antioxidants status in aged SWFs induced by D-gal. Different lowercase letters indicate significant differences (*p* < 0.05).

**Figure 6 cells-13-00415-f006:**
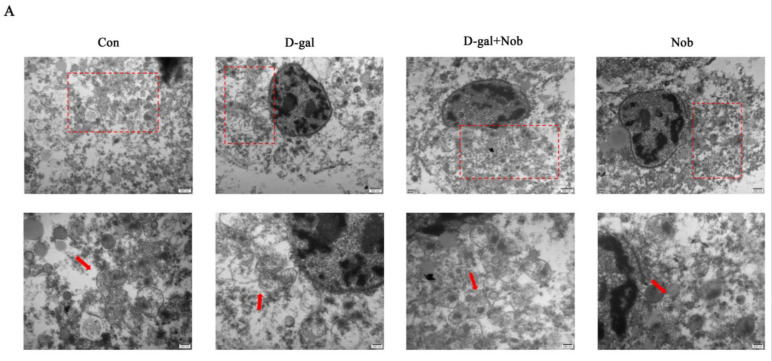
Activation of AMPK and SIRT1 signaling pathways is responsible for mitophagy induced by Nob, which protects the aged SWF−GCs. (**A**) Mitochondrial swelling was observed after D−gal treatment and was reduced with Nob. The box indicates the enlarged area below. Scale bars: 1 µm and 500 nm. (**B**–**J**) Western blot and quantitative analyses of p-AMPK/AMPK, SIRT1, FOXO3a, LC3-II, PINK1, Parkin, and NF−κB proteins in control, D−gal, D−gal + Nob, and Nob groups, β−actin as the control. (**K**) ATP levels in SWFs that underwent the treatments mentioned above (**B**–**J**). Different lowercase letters indicate significant differences (*p* < 0.05).

**Figure 7 cells-13-00415-f007:**
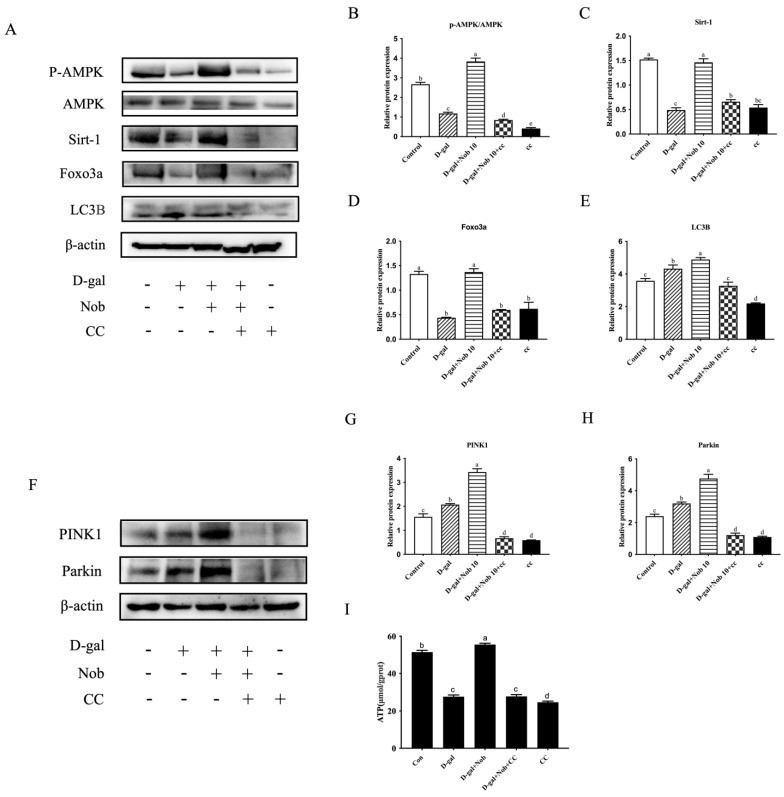
AMPK/SIRT1 signaling pathways are involved in Nob-induced mitophagy in senescent SWF−GCs induced by D−gal. (**A**–**E**) AMPK, AMPK phosphorylation, SIRT1, FOXO3a, and LC3B were quantitatively analyzed using Western blots in the control, D−gal, D−gal + Nob, D−gal + Nob + CC, and CC alone groups. (**F**–**H**) PINK1 and Parkin were quantitatively analyzed using Western blots in the control, D−gal, D−gal + Nob, D−gal + Nob + CC, and CC alone groups. (**I**) ATP content of SWFs. (**J**–**N**) Expression levels of AMPK, AMPK phosphorylation, SIRT1, FOXO3a, and LC3B in control, D−gal, D−gal + Nob, D−gal + Nob + EX−527, and EX−527 alone groups were quantitatively analyzed using Western blots, β-actin as the control. (**O**–**Q**) Expression levels of PINK1 and Parkin in the control, D−gal, D−gal + Nob, D−gal + Nob + EX−527, and EX−527 alone groups were determined using Western blotting, β-actin as the control. (**R**) ATP content of SWFs. Different lowercase letters indicate statistically significant differences (*p* < 0.05).

**Figure 8 cells-13-00415-f008:**
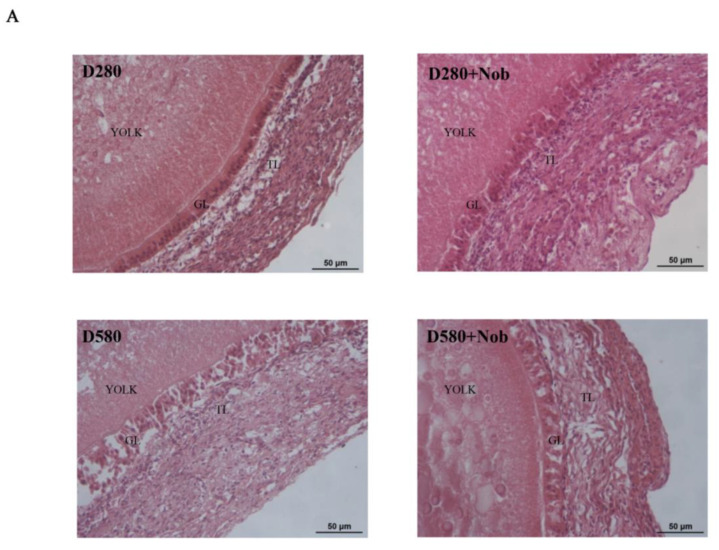
Effects of Nob-activated mitochondrial autophagy on anti-inflammatory ability, cell proliferation, and apoptosis in in vitro cultured SWFs of D280 and D580 laying chickens. (**A**) Representative H&E staining of SWFs from high- and low-producing laying chickens. Scale bar: 50 µm. (**B**–**E**) D280 and D580 chicken SWFs were incubated with BrdU (red) and labeled with TUNEL (green) as indicated. The nuclei were stained with DAPI (blue). Scale bar: 50 µm. (**F**–**I**) Effect of Nob on protein expression of PINK1, Parkin, and NF-κB in D280 and D580 chicken SWFs in vitro, β-actin as the control. Different lowercase letters indicate statistically significant differences (*p* < 0.05). (**J**–**M**) In vitro, the effect of Nob on cell proliferation and apoptosis-related protein expression in the SWFs of D280 and D580 chickens. Data are expressed as the means ± s.e. Ns, the difference between the two groups was not statistically significant. ** *p* < 0.01, *** *p* < 0.001, **** *p* < 0.0001.

**Figure 9 cells-13-00415-f009:**
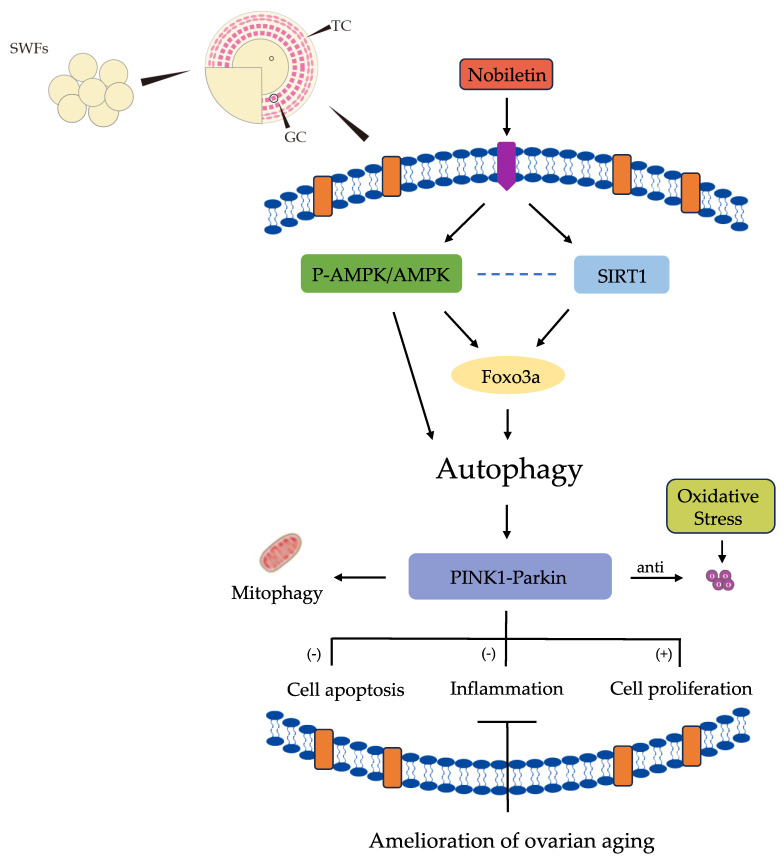
Nob has a protective impact on the GCs induced by D−gal in prematurely aging chickens. Nob regulates D−gal−induced cell phagocytosis and mitophagy in prematurely aging GCs by activating AMPK/SIRT1–FOXO3a signaling pathway. Therefore, Nob plays an anti−oxidative stress and anti−inflammatory role and alleviates ovarian aging.

**Table 1 cells-13-00415-t001:** Primers for PCR analysis.

Genes	Accession No.	Primer Sequence (5′–3′)
*CDK-2*	NM_001199857.1	TCCGTATCTTCCGCACGTTG GCTTGTTGGGATCTAGTGC
*PCNA*	NM_204170.2	GGGCGTCAACTAAACAGCA AGCCAACGTATCCGCATTGT
*CCND1*	NM_205381.1	CCGAAGGTTGTGTTCCAGTGAGAG CGTGTGTTGGCACCAAAGGATTTC
Caspase-3	NM_204725.2	ATTGAAGCAGACAGTGGACCAGATG TGCGTTCCTCCAGGAGTAGTAGC
*Bcl-2*	NM_205339.2	GCTGCTTTACTCTTGGGGGT CTTCAGCACTATCTCGCGGT
*CAT*	NM_001031215.2	CGGGATGCAATGTTGTTTCCAT AGACTCAGGGCGAAGACTCA
*SOD1*	NM_205064.2	TGACCTCGGCAATGTGACTG CACTTTTTGCATGGACCACCA
β-actin	NM_205518	ACACCCACACCCCTGTGATGAA TGCTGCTGACACCTTCACCATTC

## Data Availability

Data are included in the article.

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
