# Peer review of "Nobiletin Ameliorates Aging of Chicken Ovarian Prehierarchical Follicles by Suppressing Oxidative Stress and Promoting Autophagy"

_cells, 2024, doi:10.3390/cells13050415_

Round 1

Reviewer 1 Report

Comments and Suggestions for Authors

This paper shows experimentally that nobiletin can delay ovarian aging in laying hens by activating the AMPK/SIRT1-FOXO3a pathway.  The authors have conducted very detailed experiments, correctly interpreted the data obtained from the experiments, and expressed the results in writing. I consider it a very valuable paper.

However, there are some points that concern me, and I indicate them below. I would appreciate your explanation or improvement.

I believe that the activation of caspase-3 is caused by the cleavage of procaspase-3 to generate cleaved caspase-3. There is no clear statement as to whether the caspase-3 band in Fig. 4 indicates procaspase-3 or cleaved caspase-3. If the band shown is a band of procaspase-3, I assume that the band would be less expressed upon caspase-3 activation. I would like to have a clear explanation of whether the band is derived from procaspase-3 or cleaved caspase-3.

Fig. 3 and Fig. 5 show that D-gal treatment increases caspase-3 mRNA expression; is caspase-3 gene transcription increased during apoptosis?  I believe that the transcript level of the caspase-3 gene remains unchanged even during apoptosis. I would appreciate a clear discussion of this as well.

The authors note that each Western blot was analyzed in triplicate experiments. Are those experiments independent experiments? In other words, are the proteins prepared and analyzed in every three experiments? The authors have provided the raw data as SUPPLEMENT DATA, but if they can show the results of the three experiments, I would like to see the results of each Western blot. In particular, for caspase-3, it would be appreciated if the authors could show the results so that we can see both procaspase-3 and cleaved caspase-3.

Line 117.

Nobiletin was dissolved in DMEM medium.  Is nobiletin easily dissolved in the medium?

The specificity of all primary antibodies used to the chicken protein is not a problem?

In the first half of the Discussion, the authors use Nob, but in the second half of the Discussion, the authors use nobiletin.

I hope these my comments are helpful.

Reviewer 2 Report

Comments and Suggestions for Authors

The manuscript describes the differences in small white follicles (SWFs) between high-producing (D280) and senescent (D580) hens and the effect of a citrus fruit-derived antioxidant, Nobiletin (Nob), on SWFs functions. Moreover, the Authors described the SWF ageing experimental model induced by D-galactose (D-gal) and the elimination of this effect with Nob treatment. The obtained results may have implications for improving the laying ability of hens.

The article is interesting, but some points should be completed before publication.

Major:

Lines 108-110: This section describes how the collected tissues were stored, but there is no information about those intended for in vitro culture.

The "Materials and Methods" chapter should clearly describe the material and methods used in the experiments. However, the description includes phrases like: “diluted as required by the experimental design” (line 119) or “unless specified otherwise ..” (line 175) and there is no explanation for that.

In the "2.2 Organ culture and treatments" paragraph it is written that the SWFs were incubated in ice-cold DMEM and later stored at 38.5°C. This is not clear and I would like to ask the Authors to explain why there is conflicting information on lines 113 and 120. Moreover, I think that the use of the term "organ" is not appropriate concerning SWF. This paragraph should explain why SWFs were incubated with bromodeoxyuridine (BrdU), dorsomorphin (CC) and selisistat. Additionally, CC and EX-527 are used in the description of the results, whereas there is no information about EX-527 in the description of the methodology. An explanation should be provided. Were these reagents specific to the poultry?

On what basis were the tested concentrations of Nob 1, 10 and 100 µg/ml selected?

The description of "2.5 Western Blot Analysis" lacks information about the percentage of skimmed milk used for blocking and information about the secondary antibodies used in the experiment.

2.6 RNA Extraction and Quantitative Real-Time PCR (qPCR)”: There is no information on what reference genes were used and what method was used to calculate the expression of the tested genes.

How many animals were used in each group? How many biological and technical replicates were used in each experiment?

The word “separated” in paragraph "3.1 Morphological, proliferative, apoptotic, and laying performance alterations associated with aging in SWFs" is misleading. In the D280 the theca layer (TL) and granule layer (GL) were distinguishable whereas these two layers were separated in the case of the D580.

I think it would be good to describe what hierarchical follicles mean (line 243 and Figure 1).

In the paragraph "3.2 Establishment of the SWFs Aging Model" (lines 259-261) the Authors provide that D280 SWFs exposed to 10mg/ml D-gal were comparable to D580 SWFs but nowhere is such a comparison demonstrated. On what basis was the comparison made?

Lines 546-547: Sentence: “SIRT1, PCNA, CCND1, and Bcl-2 expression decreases, while BAX, Caspase-3, and NF-κB expression increases.” is unclear. In what cases does this expression increase?

Minor:

Line 78: the abbreviation for Nobiletin should be entered on first use, not in material and methods (line 117). The entire manuscript should be edited for the use of all abbreviations. They should be explained upon first use and applied consistently after that. Moreover, they should be the same throughout the manuscript, including figures.

At the end of the description of Figure 6, there is a fragment of the previous paragraph.

Line 212: “The” instead of “the”

Line 638: “mitochondrial”

Reviewer 3 Report

Comments and Suggestions for Authors

he aim of the study is very interesting dealing with knowing and in the future delaying chicken peripheral ovarian follicular aging. The study is nicely designed. Analyses are performed on both microscopic and molecular levels. Rich data are obtained on apoptosis, senescence autophagy, and Nobiletin treatment.

1. chicken or hen-please consider

2. It will be nice to have full chemical characteristics of Nobiletin; together with its metabolism, molecular mechanisms of action etc

3. Describe the treatment more clearly; maybe subsections are important to add

4. How were treatment doses calculated

5. Sometimes microphotographic documentation is too small

6. Provide study limitations

Round 2

Reviewer 1 Report

Comments and Suggestions for Authors

This revised version addresses my comments well.

Author Response

We appreciate the reviewer's comments.

Reviewer 2 Report

Comments and Suggestions for Authors

Lines 189-190:  Information about which protein specifically was used to normalize the Western Blot results in the manuscript's main text is still unclear. The authors explained it more precisely in their response, but I think this information should be included in the manuscript.

I still don't understand what the information that more than 5 animals and three biological and technical replicates means. How many chickens, how many SWFs and how many samples (RNA, protein, etc.) were used in each experiment? More than 5 indicates that there may have been an unequal number of results obtained in the compared groups. If so, it is not justified to use one-way ANOVA for statistical analysis. The individual categories of the independent variable should be statistically equal.

Reviewer 3 Report

Comments and Suggestions for Authors

My comments were involved in the correction. Please only provide study limitations at the end of the Discussion.
